# Financial and Other Life Stressors, Psychological Distress, and Food and Beverage Consumption among Students Attending a Large California State University during the COVID-19 Pandemic

**DOI:** 10.3390/ijerph20043668

**Published:** 2023-02-18

**Authors:** Paulina Lin, Kathryn Hillstrom, Kimberly Gottesman, Yuane Jia, Tony Kuo, Brenda Robles

**Affiliations:** 1Department of Epidemiology, UCLA Fielding School of Public Health, P.O. Box 951722, Los Angeles, CA 90095, USA; 2Department of Nutrition and Food Science, California State University, Los Angeles, 5151 State University Drive, Los Angeles, CA 90032, USA; 3School of Health Professions, Rutgers Biomedical and Health Sciences, 65 Bergen St., Newark, NJ 07101, USA; 4Department of Family Medicine, David Geffen School of Medicine at UCLA, 10880 Wilshire Blvd, Suite 1800, Los Angeles, CA 90024, USA; 5Population Health Program, UCLA Clinical and Translational Science Institute, 10833 Le Conte Ave., BE-144 CHS, Los Angeles, CA 90095, USA; 6Research Group on Statistics, Econometrics, and Health (GRECS), University of Girona, Carrer de la Universitat de Girona 10, Campus de Montilivi, 17003 Girona, Spain

**Keywords:** financial and other life stressors, mental health, psychological distress, dietary behaviors, college students, COVID-19

## Abstract

The novel coronavirus disease 2019 (COVID-19) pandemic abruptly disrupted the daily lives and health of college students across the United States. This study investigated several stressors (e.g., financial strain/uncertainty), psychological distress, and dietary behaviors among college students attending a large state university during the pandemic. A cross-sectional online survey was administered to students from the California State University, Los Angeles between April and May 2021 (final analytic sample n = 736). Differences in gender and race/ethnicity were examined using chi-square, *t*-test, and one-way ANOVA tests. Paired t-tests were performed to compare variables before and during the pandemic. Negative binomial regression models examined the associations between various stressors, psychological distress, and three key dietary outcomes. Descriptive results showed that the consumption of fruits and vegetables, fast food, and sugary beverages, along with psychological distress, all increased during the pandemic. Significant differences in fruit and vegetable and fast food consumption by gender and race/ethnicity were also observed. In the regression models, several stressors, including financial strain and psychological distress, were associated with unfavorable food and beverage consumption, thereby suggesting that college students may need more support in mitigating these stressors so they do not manifest as poor dietary behaviors. Poor diet quality is associated with poor physical health outcomes such as premature development of type 2 diabetes or hypertension.

## 1. Introduction

The novel coronavirus disease 2019 (COVID-19) pandemic undeniably disrupted the lives of college students around the globe. The advent of this historic event and its prolonged progression has taken a significant toll on both the psychology and the behavioral health of youth and young adults, manifesting, frequently, as increases in poor dietary and other lifestyle choices [1,2,3]. These choices include unhealthy behaviors such as excess alcohol consumption [4,5,6], frequent binge eating [7,8], and the overconsumption of ultra-processed foods [9]. Such behaviors may have long-term consequences beyond just exposure to and infection with the SARS-CoV-2 virus, the agent that causes COVID-19. Although less well understood and characterized in the literature, the deterioration of this group’s overall diet quality due to pandemic-related stressors, including poor mental health, is particularly concerning. After all, poor diet quality and related lifestyle choices, in general, are linked to the premature development of chronic conditions such as type 2 diabetes [10], hypertension [11], and heart disease later in life [12].

During the pandemic, poor diet quality was likely the result of or was associated with the cumulative stresses that college students experienced during the global emergency. Notable pandemic-related stressors, which have been highlighted in the literature, include financial hardship and uncertainty [13,14], psychological distress from contracting the disease itself or related bereavement [15], and social isolation amplified by lockdowns [16]. Unsurprisingly, there is ample evidence to suggest that this public health emergency has negatively impacted college students’ mental health [13,14,15,16,17,18] and that these declines in psychological well-being have exacerbated emotional eating in this group [19,20], similar to, if not worse than, the patterns observed for the general population [21,22]. Some investigators have suggested that many of these deleterious consequences, especially those experienced by college students, were in part due to a worsening or widening of sociodemographic disparities that were already prevalent before the global emergency [23,24]. However, to the best of our knowledge, few studies have described or examined these and other life stressors which may have promoted unhealthy food and beverage consumption among college students during the pandemic.

To address this gap in student health promotion and practice, the present cross-sectional study assessed the potential associations between financial and other life stressors, psychological distress, and dietary behaviors among college students who attended one of the largest four-year, United States (U.S.) public state universities during the pandemic. Study analyses were guided by three main objectives: (i) to document gender and racial/ethnic differences in these college students’ pandemic-related stressors, including psychological well-being, and their dietary behaviors; (ii) to describe perceived changes in these stressors and dietary behaviors over time, from before and during the pandemic; and (iii) to determine which of these stressors are associated with the three modifiable dietary behaviors of interest: fruit and vegetable, fast food, and soda and other sugar-sweetened beverage consumption.

## 2. Materials and Methods

### 2.1. Study Population

In mid-March 2020, California became the first state to mandate an executive stay-at-home order seeking to curb the spread of COVID-19 [25]. This order, alongside corresponding state and local-level policies [26], led California State University (CSU) to transition from in-person classes to remote learning for the Spring 2020 semester [27]. Due to the unmitigated nature of the pandemic, remote learning continued for the remainder of the academic school year across all CSU campuses and into 2021 and the start of 2022 [28]. Students in the present study were selected from the California State University, Los Angeles (Cal State LA)—among one of the CSU campuses federally designated as a “Hispanic-Serving Institution, Minority-Serving Institution, and Asian American and Native American Pacific-Islander Serving Institution” [29]. Table 1 shows the demographic characteristics and class standing of the survey sample, compared with the general Cal State LA population based on the 2021 spring enrollment statistics and information from the *Cal State LA 2020 Facts Sheet* [30]. Most of the participants in the present survey were women (78.3%) and self-identified as Hispanic (64.7%), followed by Asian and Pacific Islander (API) (19.4%), White (10.1%), and Indigenous (5.8%). From the *Cal State LA 2020 Facts Sheet*, there were more women (59.6%) than men (40.3%). The majority identified as Hispanic (69.4%), followed by API (11.7%), White (5.4%), and African American (3.6%). The 2021 Spring Enrollment Statistics (31) showed similar distribution across some of these groups, in particular Hispanics.

### 2.2. Study Design and Survey Instrument

A cross-sectional, web-based survey was developed by Cal State LA faculty and students, in collaboration with evaluators from the Los Angeles County Department of Public Health. Wherever possible, questions from validated questionnaires were used. Some questions were informed by previous surveys distributed by the CSU system [31] and those developed and previously used by the Los Angeles County Department of Public Health [32,33,34]. On average, the survey took participants 15 minutes to complete. Data were collected between April and May 2021, which was about a year after California issued its stay-at-home order and after the Cal State LA campus had shifted its instruction to remote learning. The survey was distributed through Survey Monkey, a commonly used online survey platform [35]. Students were invited to complete the Cal State LA *Health During the COVID-19 Pandemic Student Survey*. The survey included sections on food security, food and beverage consumption, sodium consumption health-related knowledge, physical activity, sleep, COVID-19 testing and diagnosis, life stressors such as financial strain/uncertainty, current health status, and sociodemographic characteristics.

### 2.3. Survey Participant Eligibility and Recruitment

Cal State LA staff involved in the present study invited all university students in the campus’ email ListServ to participate in this voluntary, anonymous, web-based survey. Prospective survey participants were sent an email invitation in April 2021 (initial email invite) and in May 2021 (follow-up email invite) by the Principal Investigator of the study. To be eligible, participants had to be enrolled as Cal State LA students at the time they received the email invitation, have successfully received the email invitation to their official Cal State LA email (i.e., no bounce backs), had to be 18 years and older, and could complete the survey in English. Student emails were not linked to their survey responses; only those who reached the end of the survey were informed that the first 100 students to complete the survey would be eligible to receive a $5 electronic gift card and were subsequently allowed to provide their preferred email addresses (in a separate section). The survey participation rate was ~5% (1299 participants clicked on the Survey Monkey invitation link/27,018 students who were emailed the invitation link). This participation rate is comparable to other web-based surveys carried out among other U.S. university campuses [36]. The calculated completion rate for the survey was 75.2%. All study protocols and materials were approved by the Institutional Review Boards at Cal State LA and the Los Angeles County Department of Public Health. Data were managed (including storage) using secured hardware and other security measures such as requiring password protection. Individuals who did not answer all of the key questions in the survey were excluded. Duplicate entries were removed from the dataset to generate a final analytic sample of 736.

### 2.4. Financial and Other Life Stressor Variables

Survey participants were asked to indicate how the pandemic impacted different aspects of their life (stressors). These questions aligned with those included in a survey of CSU students’ basic needs [37]. Participants were asked a series of multiple-choice questions, with some utilizing a Likert scale. This included asking participants, “Has your work income changed due to COVID-19 pandemic?” and “Have your expenses changed due to the COVID-19 pandemic [e.g., cost of moving back home, transportation, breaking your lease, etc.]?” Response options to these questions were shortened to categories of “Decrease” (coded as 1), “Increase” (coded as 2), and “Stayed the same” (coded as 3) for data analysis purposes. A question on financial support asked “During COVID-19 pandemic, how often have you provided financial support to loved ones, family, and/or others?” with the answer options “Always,” “Most of the time,” “About half of the time,” “Sometimes,” and “Never.” Based on the distribution of the responses, this variable was collapsed into three categories: “Majority of the time” (Always/Most of the time/About half of the time) (coded as 1), “Sometimes” (coded as 2), and “Never” (coded as 3). Participants were also asked a yes (coded as 1) or no (coded as 2) question to indicate if their housing status changed (“Did you have to return to live with parents/family due to the COVID-19 pandemic?”). Lastly, participants were asked “How worried are you that you, or your loved ones, may contract COVID-19?” with the answer options “Extremely worried,” “Somewhat worried,” “A little worried,” “Not worried at all,” and “Not applicable”. This variable was collapsed into three categories: “Extremely worried” (coded as 1), “Somewhat/a little worried” (coded as 2), and “Not worried/not applicable” (coded as 3).

### 2.5. Psychological Well-Being Variable as Measured by Level of Distress

#### Psychological Distress

Psychological distress, which is defined by or refers to symptoms of stress, anxiety, and depression [38], was measured using questions adapted from the PHQ-2 [39] and GAD-2 [40]. These instruments have been found to have clinical validity, including good test-retest validity (PHQ-2 = 0.79, GAD-2 = 0.81) and internal consistency (PHQ-2 = α = 0.81; GAD-2 = α = 0.77) [41]. For example, these instruments are frequently used to screen for depression and anxiety, respectively, in real-world settings. Participants were asked to report how often they were typically bothered by the following problems both “before the pandemic” and “over the last 2 weeks”: (a) “little interest or pleasure in doing things” (PHQ-2); (b) “feeling down, depressed or hopeless” (PHQ-2); (c) “feeling nervous, anxious or on edge” (GAD-2); and (d) “not being able to stop or control worrying” (GAD-2). For all of these questions, participants could choose “Nearly everyday,” “More than half of the days,” “Several days,” or “Not at all.” To score these items according to the PHQ-2 [42] and GAD-2 [43] scales, the responses were converted to numeric values (3 = Nearly everyday, 2 = More than half of the days, 1 = Several days, 0 = Not at all).

The responses from the PHQ-2 questions were summed to generate a final PHQ-2 score ranging from 0 to 6, with a higher score indicating potential depression. The same was done for the GAD-2 questions; higher scores indicated potential anxiety. To generate an overall measure of psychological distress, the final PHQ-2 and GAD-2 scores were summed to create a PHQ-4 score, which has been used as a screening tool for both anxiety and depression [44]. Continuous variables to represent changes in PHQ-4 scores were created by subtracting the value for before the COVID-19 pandemic from over the last 2 weeks and used in all analyses. 

### 2.6. Dietary Behavior Variables

#### 2.6.1. Fruit and Vegetable Consumption

Survey participants were asked to indicate their daily consumption levels of fruits and vegetables both “before the COVID-19 pandemic” and “during the past month [during the pandemic].” These questions were informed by evidence that consuming fruits and vegetables plays an important role in protecting a person against chronic diseases [45]; they were adapted from previous public health studies [32,33]. For fruit consumption, participants were asked, “In an average day [before the COVID-19 pandemic began/during the past month], about how many servings of fruit did you eat, counting fresh, canned, dried or frozen fruits? A serving is defined as the following: (a) 1 medium fruit (such as apples, oranges, bananas, pears); (b) ½ cup chopped, cooked, or raw fruit; or (c) ¾ cup fruit juice”. For vegetable consumption, participants were asked, “In an average day [before the COVID-19 pandemic began/during the past month], about how many servings of vegetables did you eat, counting fresh, canned, and dried frozen vegetables? A serving is defined as the following: (a) 1 cup of raw leafy vegetables (such as lettuce); (b) ½ cup of other vegetables (either chopped, cooked, or raw); or (c) ¾ cup of vegetable juice”.

Responses were reported as whole number values. The fruit and vegetable values were combined for a total consumption measure and analyzed as a continuous variable. A variable to represent changes in fruit and vegetable consumption was calculated by subtracting the value for before the COVID-19 pandemic from in the past month. This difference was converted to a categorical variable for analysis, with values greater than zero categorized as “Increase” (coded as 1), values less than zero categorized as “Decrease” (coded as 2), and values equal to zero categorized as “No change” (coded as 3).

#### 2.6.2. Fast Food Consumption

Fast food consumption was an item of interest in this study because this behavior is associated with greater intake of energy-dense and nutrient-poor foods, as well as an increased risk for chronic conditions [46]. Adapted from questions used in previous public health studies [32,34], participants were asked, “In an average week [before the COVID-19 pandemic began/during the past month], about how many meals did you purchase from a fast food, sit-down, or similar restaurant (e.g., McDonald’s, Denny’s, food trucks/carts, etc.) did you eat? This includes food eaten inside the restaurant, outside the restaurant (e.g., outdoor restaurant patio), or purchased for take-out (e.g., Grubhub, Postmates, DoorDash, Uber Eats, etc.)”. Responses were reported as whole number values and analyzed as a continuous variable. A change in the fast food consumption variable was constructed by subtracting the value for before the COVID-19 pandemic from in the past month and converted to a categorical variable for analysis. Positive values were categorized as “Increase” (coded as 1), negative values as “Decrease” (coded as 2), and values of zero as “No change” (coded as 3).

#### 2.6.3. Sugary Beverage Consumption

Consumption of sugary beverages was also included in the study because of their association with chronic conditions (e.g., diabetes, metabolic syndrome) [47]. These questions were also adapted from previous public health studies [32,33,34]. Participants were asked, “In an average week [before the COVID-19 pandemic began/during the past month], about how many regular sodas such as Coke or Mountain Dew, did you drink? Do not include diet sodas or sugar-free drinks. Please count a 12-ounce can, bottle, or glass as one drink”. They were also asked, “In an average week [before the COVID-19 pandemic began/during the past month], how many other sugar-sweetened beverages [SSBs] did you drink? SSBs include sports or energy drinks such as Gatorade, Powerade, or Red Bull (do not include sugar-free versions of these drinks). SSBs can also include sweetened drinks such as Kool-Aid, lemonade, coffee, and other drinks you made at home and added sugar to”. Responses to each question were reported as a whole numeric number.

These two values were combined for a total sugary beverage consumption measure and analyzed as a continuous variable. A variable to represent changes in consumption was constructed by subtracting the value for before the COVID-19 pandemic from in the past month. This was converted to a categorical variable for analysis, with positive values categorized as “Increase” (coded as 1), negative values as “Decrease” (coded as 2), and values equal to zero as “No change” (coded as 3).

### 2.7. Covariates

Several questions about gender, age, race and ethnicity, marital status, number of dependents, and class standing were included in the survey. Participants also answered how much money they have in comparison with others of the same age, a question that has been used in prior studies to assess relative financial strain in younger populations [48]. For race/ethnicity, participants were given the option to check more than one race when answering, which were then categorized as White, Hispanic, API, and Indigenous. For statistical modeling purposes, only the race and ethnicity categories with greater than 5% frequencies were retained. For instance, students who identified as Black were excluded from the analyses due to a small sample size.

### 2.8. Statistical Analyses

A conceptual model (not shown) based on pre-specified hypotheses, and informed by empirical literature, guided all study analyses. Descriptive analyses were initially generated to describe the demographic characteristics and class standing of the participants, displaying percentages for categorical variables and means and standard deviations for continuous variables. Differences in financial and other life stressors, psychological distress, and dietary behaviors were examined by gender and race/ethnicity. These comparisons were assessed using chi-square tests, t-tests, and one-way ANOVA (two-tailed) tests. Paired t-tests were used to evaluate for statistically significant changes in the level of psychological distress and dietary behaviors among college students before and during the COVID-19 pandemic. Models using negative binomial regressions were constructed to examine the associations between several stressors and the three dietary behaviors of interest (i.e., fruit and vegetable, fast food, and sugary beverage consumption); all models were controlled for covariates. All analyses were conducted using the SAS 9.4 statistical software package (SAS Institute, Inc., Cary, NC, USA). A *p*-value of <0.05 was considered statistically significant.

## 3. Results

### 3.1. Objective 1: To Document Gender and Racial/Ethnic Differences in College Students’ Pandemic-Related Stressors, Including Psychological Well-Being, and Their Dietary Behaviors

Differences in survey participants’ financial and other pandemic-related stressors, including psychological well-being, and their dietary behaviors, by gender and race/ethnicity are presented in Table 2.

In the full sample, most students who participated in the survey indicated that their income and expenses (48.5% and 45.2%, respectively) stayed the same during the COVID-19 pandemic; 74.5% indicated they had not returned to live with their parents/family. About a third (37.0%) reported they were extremely worried about contracting COVID-19 and that they provided financial support to their loved ones a ‘majority of the time’ (32.3%). Almost a quarter (24.9%) disclosed they had accessed a campus basic needs program. In subgroup comparisons, several statistically significant gender differences were observed in the analyses, including a greater decrease in work income due to the pandemic for students who identified as a woman versus students who identified as a man (*p* = 0.0133). Furthermore, more students who identified as a woman (versus a man) indicated they were worried about contracting COVID-19 (*p* = 0.028). Several racial/ethnic differences were noted in the analyses. For example, compared with White students, more Hispanic students provided financial support to their loved ones (*p* = 0.0014); more Hispanic and Indigenous students were worried about contracting COVID-19 (*p* = 0.0009); and more Indigenous students accessed a campus basic needs program (*p* = 0.0057).

In the full sample, the overall mean PHQ-4 score (i.e., a measure of psychological distress) increased by 1.5 points from before to during the COVID-19 pandemic. In subgroup comparisons, several statistically significant gender and racial/ethnic differences in the scores were seen. For instance, before the COVID-19 pandemic, students (survey participants) who identified as a woman had, on average, a 0.6 higher PHQ-4 score than students who identified as a man (*p* = 0.456). In contrast, during the pandemic, students who identified as a woman had, on average, a 1.7 higher PHQ-4 score than students who identified as a man (*p* ≤ 0.0001). For race/ethnicity, one of the more notable findings was that Indigenous students had the highest average PHQ-4 scores (6.2) than other subgroups—White (5.5), Hispanic (5.7), API (4.6)—during the pandemic (*p* = 0.0216).

In the full sample, from before to during the COVID-19 pandemic, the average number of fruits and vegetables consumed increased by 0.7 daily servings and the number of fast food meals increased by 0.3 meals per week and sugary beverages by 0.2 beverages per week. There were a few differences in sugary beverage consumption by race/ethnicity. In particular, White students consumed the lowest average number of sugary beverages before the pandemic (2.9) compared with students who were Hispanic (4.3), API (3.2), and Indigenous (5.6) (*p* = 0.0006). In contrast, during the pandemic (i.e., “in the past month”), API students consumed the lowest average number of sugary beverages (3.5), compared with White (3.8), Hispanic (4.3), and Indigenous (7.0) students (*p* = 0.0038).

### 3.2. Objective 2: To Describe Perceived Changes in College Students’ Pandemic-Related Stressors and Dietary Behaviors over Time, from before and during the Pandemic

Perceived changes in pandemic-related stressors, in particular psychological well-being, and food and beverage consumption by gender and race/ethnicity are presented in Figure 1.

Except for students who identified as API, psychological distress significantly increased for all groups from before to during the pandemic. PHQ-4 scores were significantly higher during the pandemic (compared with before the pandemic) in the full sample (*p* < 0.001) and among students (survey participants) who identified as a woman (*p* < 0.05), as a man (*p* < 0.001), as White (*p* < 0.01), as Hispanic (*p* < 0.001), and as Indigenous (*p* < 0.01).

Overall, food consumption among students significantly increased from before to during the pandemic. Statistically significant increases in the mean fruits and vegetables consumed in a day were observed in the full sample (*p* < 0.01) and students (survey participants) who identified as a woman (*p* < 0.01), as a man (*p* < 0.05), and Hispanic (*p* < 0.001). Similarly, statistically significant increases in the mean number of fast food meals consumed in a week were observed in the full sample (*p* < 0.05) and among students who identified as a woman (*p* < 0.05) and as Hispanic (*p* < 0.05). Increases in the number of sugary beverages consumed in a week were not statistically significant in the analyses.

### 3.3. Objective 3: To Determine Which of the Pandemic-Related Stressors Are Associated with the Three Dietary Behaviors of Interest: Fruit and Vegetable, Fast Food, and Sugary Beverage Consumption

Perceived changes in pandemic-related stressors, in particular psychological well-being, by gender and race/ethnicity are also presented in Figure 1. The results from the negative binomial regression models are presented in Table 3.

#### 3.3.1. Fruit and Vegetable Consumption

Among survey participants, students who reported a decrease in income consumed 13% less servings of fruits and vegetables than students whose income remained the same (Incidence Rate Ratio [IRR] = 0.87, 95% Confidence Interval [CI] = 0.76, 0.98), holding the other variables constant. Similarly, students who reported an increase in expenses consumed 17% less servings of fruits and vegetables than students whose expenses remained the same (IRR = 0.83, 95% CI = 0.72, 0.95), holding the other variables constant. Compared with those who never had to provide financial support to their family/loved ones, students who provided such support for a “majority of the time” and “sometimes” during the pandemic consumed 34% and 19% more servings of fruits and vegetables, respectively, (IRR = 1.34, CI = 1.15, 1.57; IRR = 1.19, CI = 1.03, 1.37) and students who returned to live with their parents or family members during the pandemic consumed 15% fewer servings of fruits and vegetables than students who did not return home (IRR = 0.85, CI = 0.74, 0.97). In these models, the covariate, class standing, was associated with fruit and vegetable consumption, with ‘third years’ consuming 22% less than ‘first years’ (IRR = 0.78, CI = 0.64, 0.96).

#### 3.3.2. Fast Food Consumption

Survey participants who provided financial support to family/loved ones a “majority of the time” and “sometimes” during the pandemic consumed 44% and 35% more fast food, respectively, than students who never had to provide such support (IRR = 1.44, CI = 1.20, 1.74; IRR = 1.35, CI = 1.14, 1.60). In the regression models, some covariates were significantly associated with weekly fast food consumption. One of them was psychological distress, which was marginally associated with higher fast food consumption (IRR = 1.04, CI = 1.02, 1.06). Another was class standing, which showed that students who are graduate/other students consumed 24% less fast food than ‘first years’ (IRR = 0.76, CI = 0.60, 0.97).

#### 3.3.3. Sugary Beverage Consumption

None of the stressors examined in the models were significantly associated with sugary beverage consumption (Table 3). However, a few covariates were marginally associated with higher sugary beverage consumption. These include psychological distress (IRR = 1.04, CI = 1.01, 1.06) and class standing, with graduate/other students consuming 28% less sugary beverages than their ‘freshmen’ counterparts (IRR = 0.72, CI = 0.52, 0.99). For race/ethnicity, the models showed that students who identified as Indigenous consumed 63% more sugary beverages than students who identified as Hispanic (IRR = 1.63, CI = 1.11, 2.39).

## 4. Discussion

### 4.1. Main Findings

The *Health During the COVID-19 Pandemic Student Survey* is the first study of its kind to explore and describe the associations between financial and other life stressors, psychological distress, and food and beverage consumption among college students at a large U.S. state university in California during the COVID-19 pandemic. Because the emergence of this viral disease heightened both emotional duress and the real risk for hospitalization and death among those experiencing chronic conditions such as diabetes [47], this infectious agent posed a significant health threat to all groups in the general population. Specifically, youth and young adults were not exempt from experiencing these consequences [49]. Cumulatively, the effects of the pandemic exposed several gaps in college students’ life stressors, including psychological distress and diet, pointing to an urgent need to address them. Several notable findings from the study support this conclusion.

First, differences in psychological distress and the consumption of fruits and vegetables and fast food by gender and race/ethnicity suggest a widening gap in mental health conditions and unhealthy eating behaviors between groups, which, while modifiable, has largely remained unnoticed and unattended in the U.S. For instance, students who identified as a woman were found to have higher levels of psychological distress than students who identified as a man. This finding is consistent with prior research on this subject matter, which has documented worsening mental health status by gender [50] and by race/ethnicity [51] among college students during the pandemic.

The present study also found notable differences in financial and other life stressors by sociodemographic characteristics. For example, an obligation to provide financial support to family/loved ones was comparatively more prevalent among students who identified as Hispanic than among students from other racial/ethnic backgrounds. A higher percentage of students who identified as a woman indicated their work income declined during the pandemic and that they were more worried about contracting COVID-19 than their male counterparts. A recent investigation showed similar COVID-19 impacts related to these factors for Latino adults and women [52].

Based on the survey results, the negative impact of financial and other life stressors, and the corresponding decline in psychological well-being, on college students’ dietary behaviors (and to a broader extent, physical health) cannot be overlooked. During the pandemic, the consumption of sugary beverages such as sodas and other SSBs was significantly higher for Indigenous students and those who had higher levels of psychological distress. These survey results were not necessarily surprising, as other research investigations have previously shown similar patterns of sugary beverage consumption by race/ethnicity among U.S. children and adolescents [53].

Second, the overall pool of study participants (college students) experienced several changes in food and beverage consumption from before and during the COVID-19 pandemic. In particular, the consumption of fruits and vegetables increased significantly among the sampled group; the net gain in positive behaviors, however, was somewhat mixed, potentially negated by the reported increase in fast food consumption. In the literature, increased eating of any kind is not uncommon under stressful conditions or emotional duress. Studies have shown that eating and snacking behaviors increased during the earlier phases of the COVID-19 pandemic [54,55,56]. During the health crisis, some of the increased consumption of fruits and vegetables and foods rich in whole grains likely paralleled the recommendations provided by national nutritional guidelines, as people tried to adhere to these and other expert advice to keep healthy [57]. Unfortunately, stressful/emotional eating may have led many to increase their fast food consumption as well. Findings from the present study add to recent research which showed that elevated stress due to the pandemic was correlated with increased motivation to purchase and consume fast food and sweets [58], and that the pandemic had both positive and negative effects on population-level dietary behaviors [59].

Third, the COVID-19 pandemic appeared to have negatively affected the psychological well-being (as measured by level of distress) of college students. Other than for APIs, the levels of psychological distress increased across all groups in the survey sample. However, these increases cannot be attributed fully to the onset of the pandemic, as even before the health crisis, there was already ample evidence suggesting that college students are more likely to experience psychosocial stressors than any other group [60]. In other words, the pandemic did not cause these psychological/mental health issues to develop as much as it exacerbated them [61,62,63], highlighting the increasing burden of poor mental health among college students and that this population has needs that are currently not being adequately addressed on university campuses. It is important to note that while the survey found no significant changes in PHQ-4 scores among APIs, it is still possible that this group has psychological distress that is unreported or underreported since, culturally, these students are less likely to talk about anxiety and depressive symptoms due to existing stigma and attitudes towards speaking about mental health issues openly [64]. Studies of Asian Americans have found that they are less likely to utilize mental/behavioral health services [65] and are frequently less likely to seek help due to feelings of shame and wanting to save face [66].

Fourth and lastly, psychological distress—along with financial stressors created by COVID-19 lockdowns and related policy restrictions—was independently and most likely unidirectionally associated with adverse changes in food and beverage consumption among college students. This finding is consistent with those from other studies that have demonstrated similar patterns due to the pandemic [20,54,55,56,67]; namely, several investigations found correlations between increased stress and eating [56,67].

Overall, it is not surprising that having to provide financial support to family/loved ones during the pandemic generated financial strain and uncertainty/instability, which, in turn, raised the overall stress and anxiety levels of many students. This correlation between financial worries and psychological/mental health likely explains why many of the students increased their consumption of fast food and sugary beverages during the pandemic [58]. Having more free time than usual is another plausible explanation for why poor dietary behaviors developed or increased during the health crisis. For example, having more free time may have inadvertently contributed to more snacking throughout the day— i.e., a behavior which has been linked to unhealthy eating [54,55].

Increased expenses and returning to live with parents or family during the pandemic may have also produced a similar trajectory as financial strain/uncertainty in driving the increase in poor dietary behaviors among college students. An interesting aspect of these factors is that they may reflect the spread of “panic buying” during the pandemic. That is, buying and hoarding items due to widespread worries of lockdown and scarcity of goods. During the early stages of the COVID-19 pandemic, grocery and store shelves were emptied because a large number of people stocked up on basics, bottled water, and food items, including in Los Angeles County where Cal State LA is located [68]. There is evidence that people opted to purchase more nonperishable and processed foods over fresh fruits and vegetables as a result of this frenzy [69]. College students who returned home during the pandemic have reported similar buying habits, as their families prioritized purchasing foods with longer shelf lives over foods that are fresh and produce-centered [54].

### 4.2. Limitations and Strengths

The present study has several limitations. First, its cross-sectional design limits the scope of the analyses. For example, causation/causal inferences could not be made with confidence given the nature of the data collection. Second, the study sample size was relatively small, drawn from a much larger overall population of Cal State LA students. The sample size also reflects the fact that the survey results represent data only from one segment of the college student population in one state university in California. Thus, selection bias from the focused sample and the differential responses by gender and race/ethnicity may have occurred (e.g., more female than male students and more students of Hispanic background completed the survey); albeit, when compared with the literature [70] and to Cal State LA’s own enrollment/student body statistics (see Table 1), these differences were not too dissimilar.

Third, study data were primarily self-reported. Thus, there may have been social desirability, reporting, and recall biases that affected the estimation of the food and beverage servings and numbers used to measure consumption behaviors. Fourth, survey questions that asked about behavioral changes are ‘perceived’ rather than objectively measured. For dietary consumption, this can be particularly problematic, since recall errors on food-related behaviors are quite common and have been known to reduce and even invalidate estimates of fruit, vegetable, and/or sugary beverage intake [71]. Fifth, not all of the survey questions were validated or based on validated measures from the literature. However, to the extent possible/feasible, most questions were adapted from existing items used previously in studies of Los Angeles County and/or CSU populations.

Sixth, the present study utilized listwise deletion as an approach to handling missing responses in the survey. Although other approaches for handling missing data (e.g., multiple imputation and other imputation methods) are available and were considered, it was not clear whether such methods would have significantly improved the power of the analyses, given that the overall survey sample size was still relatively small at the start of the analysis. Additionally, many of these methods are known to introduce bias and can have numerical and processing problems with the algorithms used [72,73].

Finally, the associations between psychological distress and dietary behaviors were not necessarily unidirectional, as several studies have shown that dietary behaviors can be drivers of poor or worsening psychological/mental health [74].

Despite these limitations, the present study had several notable strengths as well. For example, the study offered unique insights into financial and other life stressors, psychological distress, and the dietary behaviors of a racially/ethnically diverse student population that was previously understudied. The survey presented an extraordinary opportunity to describe many of the aforementioned conditions and outcomes for specific subgroups (e.g., Indigenous students) who are underrepresented in research and have not been historically prioritized due to frequent under-sampling in population-based research.

## 5. Conclusions

The *Health During the COVID-19 Pandemic Student Survey* considered an array of factors that may have worsened the psychological and physical well-being of college students in the U.S. during the COVID-19 pandemic. The latter outcome manifested itself as poor dietary behaviors among college students who faced several financial/social conditions that arose as a result of the health crisis; diet quality alone is an important condition that has been largely overlooked in this nation’s overall response to the global emergency. In particular, financial strain and uncertainty from loss of work income and family obligations appeared to play critical roles in determining how these students adjusted and adapted to the pandemic, and how they functioned under this duress. These study findings also add to an emerging viewpoint, especially among the public, that globally and as a nation, the U.S. public health response fell short in meeting many people’s mental and physical health needs during the crisis, especially for young and older adults [75]. Further ‘post-mortem’ evaluation of this response is certainly advisable and would be needed to better prepare the U.S. and the world for future pandemics and emergencies. Such assessment should be rigorous and pay particular attention to whether public health policies and university interventions during the crisis, while well-intentioned, were actually benefitting college students. Recommendations from this evaluation should focus on providing college students with more appropriate resources and tools that can help them better manage their emotional/psychological/mental health as well as their diet and other conditions that may affect their physical health. Hopefully, out of this retrospective examination and learning, more tailored, meaningful strategies, including public awareness/risk communication campaigns, can be formulated and put in place so both U.S. college students and students from abroad will be better served the next time the world faces another global public health emergency.

## Figures and Tables

**Figure 1 ijerph-20-03668-f001:**
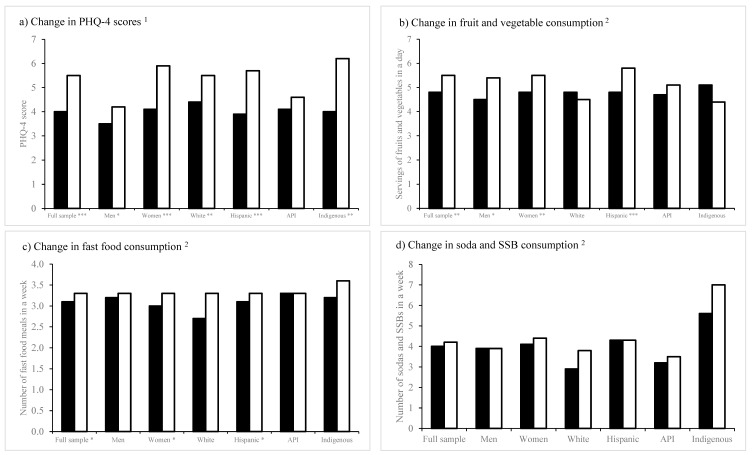
Perceived changes in food and beverage choices and mental health indicators among survey participants from the *Health During the COVID-19 Pandemic Student Survey* stratified by gender and race, California State University Los Angeles, 2021. ^1^ (**a**) legend: ■ Before the COVID-19 pandemic □ Over the last two weeks; ^2^ (**b**–**d**) legend: ■ Before the COVID-19 pandemic □ In the past month. * *p* < 0.05, ** *p* < 0.01, *** *p* < 0.0001.

**Table 1 ijerph-20-03668-t001:** Demographics of students at the California State University, Los Angeles from the 2021 *Health During the COVID-19 Pandemic Student Survey*, Cal State LA 2021 Spring Semester Enrollment Statistics, and the *Cal State LA 2020 Facts Sheet*.

Characteristics	2021 Survey	2021 Spring Enrollment Statistics ^1^	2020 Facts Sheet ^2,^*
**Total**	**736 (100%)**	**24,197 (100%)**	**26,342 (100%)**
Gender			
Women	576 (78.3%)	14,670 (59.7%)	15,701 (59.6%)
Men	160 (21.7%)	9863 (40.1%)	10,620 (40.3%)
Non-binary/Other	-- (--) ^3^	41 (0.2%)	21 (0.1%)
Age, mean	25.4	Not available	24.1
Class Standing			
Freshman	83 (11.3%)	3664 (14.9%)	4896 (18.6%)
Sophomore	63 (8.6%)	2780 (11.3%)	3477 (13.2%)
Junior	167 (22.7%)	5081 (20.7%)	5670 (21.5%)
Senior	260 (35.3%)	9632 (39.2%)	8523 (32.4%)
Graduate student and other	163 (22.2%)	3417 (13.9%)	3776 (14.3%)
Race/Ethnicity			
Hispanic	476 (64.7%)	16,674 (68.9%)	18,283 (69.4%)
Asian/Pacific Islander	143 (19.4%)	2829 (11.7%)	3076 (11.7%)
White	74 (10.1%)	1375 (5.7%)	1429 (5.4%)
Indigenous	43 (5.8%)	25 (0.1%)	26 (0.1%)
Mixed race	83 (11.3%)	352 (1.5%)	370 (1.4%)
Black/African American	-- (--) ^3^	868 (3.6%)	936 (3.6%)
Non-resident	-- (--) ^4^	1486 (6.1%)	1623 (6.2%)
Unknown	-- (--) ^4^	564 (2.3%)	599 (2.3%)

* Reported as n (%) unless otherwise specified. ^1^ Data source: California State University, Los Angeles Spring 2021 Enrollment Statistics (n = 24,197) (published data, 2021). ^2^ Data source: California State University, Los Angeles *2020 Facts Sheet* (n = 26,342) (published data, 2021). ^3^ Categories were excluded from the final analytic sample for the *Health During the COVID-19 Pandemic Students Survey* (n = 736) due to low percentages (<5%). Nineteen (2.4%) of the total participants reported they were non-binary, 31 (3.9%) of the total participants reported they were mixed race, and 19 (2.4%) participants reported they were Black/African American. ^4^ Categories were not included in the questions for the *Health During the COVID-19 Pandemic Students Survey*.

**Table 2 ijerph-20-03668-t002:** Survey participant characteristics from the *Health During the COVID-19 Pandemic Student Survey* by gender and race/ethnicity, California State University, Los Angeles, 2021.

	Full Sample	Gender			Race/Ethnicity				
Total (%)	736 (100%)	Men 160 (21.7%)	Women 576 (78.3%)	*p*-Value	White 74 (10.1%)	Hispanic 476 (64.7%)	API 143 (19.4%)	Indigenous 43 (5.8%)	*p*-Value
**Dietary Behaviors ^a^**									
Fruits and vegetables in a day (servings), mean (SD)									
Before the pandemic	4.8 [3.3]	4.5 [3.1]	4.8 [3.4]	0.3088 ^2^	4.8 [2.7]	4.8 [3.3]	4.7 [3.9]	5.1 [2.9]	0.9483 ^3^
During the pandemic	5.5 [5.6]	5.4 [5.1]	5.5 [5.7]	0.9597 ^2^	4.5 [2.9]	5.8 [6.0]	5.1 [5.9]	4.4 [2.5]	0.1373 ^3^
Change	0.7 [5.1]	0.9 [4.3]	0.6 [5.3]	0.4946 ^2^	−0.2 [2.6]	1.0 [5.5]	0.4 [5.1]	−0.7 [3.0]	0.0508 ^3^
Fast food meals in a week, mean (SD)									
Before the pandemic	3.1 [2.7]	3.2 [3.3]	3.0 [2.5]	0.4581 ^2^	2.7 [2.1]	3.1 [2.4]	3.3 [4.0]	3.2 [1.7]	0.4565 ^3^
During the pandemic	3.3 [3.5]	3.3 [3.5]	3.3 [3.5]	0.8035 ^2^	3.3 [4.5]	3.3 [3.3]	3.3 [3.8]	3.6 [3.0]	0.9758 ^3^
Change	0.3 [3.2]	0 [3.1]	0.3 [3.2]	0.3141 ^2^	0.6 [4.4]	0.3 [2.9]	0.0 [3.3]	0.4 [3.2]	0.5384 ^3^
Sugary beverages in a week, mean (SD)									
Before the pandemic	4.0 [4.6]	3.9 [4.3]	4.1 [4.6]	0.5610 ^2^	2.9 [4.0]	4.3 [4.2]	3.2 [4.4]	5.6 [7.7]	0.0006 ^3,^**
During the pandemic	4.2 [5.7]	3.9 [5.6]	4.4 [5.7]	0.3254 ^2^	3.8 [7.4]	4.3 [5.1]	3.5 [4.8]	7.0 [9.8]	0.0038 ^3,^**
Change	0.2 [4.7]	0 [4.7]	0.3 [4.7]	0.5284 ^2^	0.9 [6.8]	0 [4.4]	0.3 [4.0]	1.4 [5.7]	0.1057 ^3^
**Psychological Well-being ^b^**									
Psychological distress (PHQ-4 score)									
Before the pandemic	4.0 [3.4]	3.5 [3.2]	4.1 [3.4]	0.0456 ^2,^*	4.4 [3.8]	3.9 [3.4]	4.1 [3.2]	4.0 [3.5]	0.6190 ^3^
During the pandemic	5.5 [4.0]	4.2 [3.7]	5.9 [4.0]	<0.0001 ^2,^***	5.5 [4.0]	5.7 [4.0]	4.6 [3.8]	6.2 [4.1]	0.0216 ^3,^*
Change	1.5 [3.7]	0.7 [3.3]	1.8 [3.8]	0.0003 ^2,^**	1.1 [2.9]	1.9 [3.9]	0.5 [3.3]	2.2 [4.5]	0.0008 ^3,^**
**Financial and Other Life Stressors ^c^**									
Change in work income due to the COVID-19 pandemic									
Declined	298 (39.8%)	48 (30%)	245 (42.5%)	0.0133 ^1,^*	30 (40.5%)	185 (38.9%)	56 (39.2%)	22 (51.2%)	0.8307 ^1^
Increased	86 (11.7%)	24 (15%)	62 (10.8%)		9 (12.2%)	55 (11.6%)	17 (11.9%)	5 (11.6%)	
Stayed the same	357 (48.5%)	88 (55%)	269 (46.7%)		35 (47.3%)	236 (49.6%)	70 (49%)	16 (37.2%)	
Change in expenses due to COVID-19									
Declined	155 (21.1%)	36 (22.5%)	119 (20.7%)	0.7243 ^1^	14 (18.9%)	96 (20.2%)	35 (24.5%)	10 (23.3%)	0.7573 ^1^
Increased	248 (33.7%)	56 (35.0%)	192 (33.3%)		25 (33.8%)	168 (35.3%)	40 (28.0%)	15 (34.9%)	
Stayed the same	333 (45.2%)	68 (42.5%)	265 (46%)		35 (47.3%)	212 (44.5%)	68 (47.6%)	18 (41.9%)	
Provided financial support to loved ones, family, and/or others									
Majority of the time	238 (32.3%)	47 (29.4%)	191 (33.2%)	0.4544 ^1^	17 (23%)	170 (35.7%)	39 (27.3%)	12 (27.9%)	0.0014 ^1,^**
Sometimes	295 (40.1%)	63 (39.4%)	232 (40.3%)		27 (36.5%)	200 (42%)	50 (35.0%)	18 (41.9%)	
Never	203 (27.6%)	50 (31.3%)	153 (26.6%)		30 (40.5%)	106 (22.3%)	54 (37.8%)	13 (30.2%)	
Return to live with parents/family									
Yes	188 (25.5%)	48 (30%)	140 (24.3%)	0.144 ^1^	14 (18.9%)	117 (24.6%)	45 (31.5%)	12 (27.9%)	0.1947 ^1^
No	548 (74.5%)	112 (70%)	436 (75.7%)		60 (81.1%)	359 (75.4%)	98 (68.5%)	31 (72.1%)	
Worry of contracting COVID-19									
Extremely worried	272 (37%)	45 (28.1%)	227 (39.4%)	0.028 ^1,^*	11 (14.9%)	191 (40.1%)	50 (35.0%)	20 (46.5%)	0.0009 ^1,^**
Somewhat/a little worried	375 (51%)	91 (56.9%)	284 (49.3%)		54 (73%)	224 (47.1%)	79 (55.2%)	18 (41.9%)	
Not worried/not applicable	89 (12.1%)	24 (15%)	65 (11.3%)		9 (12.2%)	61 (12.8%)	14 (9.8%)	5 (11.6%)	
Accessed campus-based needs program									
Yes	183 (24.9%)	37 (23.1%)	146 (25.4%)	0.5651 ^1^	19 (25.7%)	115 (24.2%)	29 (20.3%)	20 (46.5%)	0.0057 ^1,^**
No/not sure	553 (75.1%)	123 (76.9%)	430 (74.7%)		55 (74.3%)	361 (75.8%)	114 (79.7%)	23 (53.5%)	

Note: API = Asian/Pacific Islander. Reported as: ^a^ mean [standard deviation], ^b^ score, or ^c^ sample size (percent). ^1^ Chi-square test. ^2^ *t*-test. ^3^ One-way ANOVA. * *p* < 0.05, ** *p* < 0.01, *** *p* < 0.0001.

**Table 3 ijerph-20-03668-t003:** The potential impact of financial and other life stressors and psychological distress on food and beverage consumption behaviors during the COVID-19 pandemic after controlling for sociodemographic characteristics: Results from the *Health During the COVID-19 Pandemic Student Survey*, California State University, Los Angeles, 2021.

Independent Variables	Model 1: Fruit and Vegetable Consumption ^1^	Model 2: Fast Food Consumption ^2^	Model 3: Sugary Beverage Consumption ^3^
	IRR (95% CI)	IRR (95% CI)	IRR (95% CI)
**Financial and Other Life Stressors**			
Change in work income (ref: Stayed the same)			
Decrease in income	0.87 (0.76, 0.98) *	1.01 (0.87, 1.17)	0.99 (0.81, 1.21)
Increase in income	1.07 (0.89, 1.28)	1.00 (0.8, 1.24)	1.07 (0.80, 1.44)
Change in expenses (ref: Stayed the same)			
Decrease in expenses	1.03 (0.89, 1.20)	0.90 (0.75, 1.07)	0.87 (0.69, 1.11)
Increase in expenses	0.83 (0.72, 0.95) **	0.98 (0.83, 1.15)	0.93 (0.75, 1.16)
Need to provide financial support to loved ones, family, and/or others (ref: Never)			
Majority of the time	1.34 (1.15, 1.57) ***	1.44 (1.20, 1.74) ***	1.18 (0.92, 1.50)
Sometimes	1.19 (1.03, 1.37) *	1.35 (1.14, 1.60) ***	1.19 (0.95, 1.49)
Returned to live with parents/family (ref: No)			
Yes	0.85 (0.74, 0.97) *	0.87 (0.75, 1.02)	0.88 (0.72, 1.08)
Worry levels of contracting COVID-19 (ref: Not worried/not applicable)			
Extremely worried	1.21 (0.99, 1.46)	1.00 (0.80, 1.25)	1.08 (0.80, 1.46)
Somewhat/a little worried	1.18 (0.98, 1.41)	1.02 (0.82, 1.26)	1.10 (0.83, 1.46)
Accessed campus-based needs programs and services (ref: No/don’t know)			
Yes	1.01 (0.89, 1.16)	0.92 (0.78, 1.08)	0.91 (0.74, 1.13)
**Covariates**			
Level of psychological distress (PHQ-4 score)	0.99 (0.97, 1.00)	1.04 (1.02, 1.06) ***	1.04 (1.01, 1.06) **
Gender (ref: Women)			
Men	1.01 (0.88, 1.16)	1.06 (0.90, 1.25)	1.03 (0.83, 1.28)
Race/ethnicity (ref: Hispanic)			
White	0.85 (0.70, 1.04)	1.07 (0.85, 1.34)	0.92 (0.67, 1.25)
Asian American/Pacific Islander	0.91 (0.79, 1.05)	1.06 (0.89, 1.26)	0.87 (0.69, 1.10)
Indigenous	0.80 (0.62, 1.02)	1.11 (0.83, 1.47)	1.63 (1.11, 2.39) *
Number of dependents	1.01 (0.97, 1.06)	1.00 (0.95, 1.06)	1.07 (1.00, 1.15)
Class level (ref: Freshman)			
Sophomore	0.88 (0.73, 1.07)	0.90 (0.72, 1.13)	1.02 (0.75, 1.37)
Junior	0.78 (0.64, 0.96) *	0.90 (0.71, 1.14)	0.85 (0.62, 1.17)
Senior	0.88 (0.73, 1.07)	0.90 (0.72, 1.13)	1.02 (0.75, 1.37)
Graduate and other students	0.82 (0.67, 1.01)	0.76 (0.60, 0.97) *	0.72 (0.52, 0.99) *
Amount of money to spend compared with others the same age (ref: Same/not sure)			
Less money	1.01 (0.88, 1.16)	0.86 (0.73, 1.01)	1.01 (0.81, 1.25)
More money	0.92 (0.78, 1.09)	0.94 (0.77, 1.15)	1.04 (0.80, 1.36)

Note: Negative binomial regressions for food and beverage outcomes, IRR = Incidence rate ratio, CI = Confidence interval. ^1^ Survey participants reported how many servings of fruits and vegetables they consumed in an average day during the past month. ^2^ Survey participants reported the number of meals they purchased from a fast food, sit-down, or similar restaurants in an average week during the past month. ^3^ Survey participants reported the number of sugary beverages (i.e., sodas and other sugar sweetened beverages) they drank in an average week during the past month. * *p* < 0.05, ** *p* < 0.01, *** *p* < 0.001.

## Data Availability

De-identified data from this study are not available in a public archive, but they can be made available upon request if the request is deemed reasonable and adheres to the study’s human subjects protection standards. Please email all inquiries to the corresponding author. Please also note that this article is original and has not been previously published in a peer-reviewed journal.

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
