# Peer review of "Financial and Other Life Stressors, Psychological Distress, and Food and Beverage Consumption among Students Attending a Large California State University during the COVID-19 Pandemic"

_ijerph, 2023, doi:10.3390/ijerph20043668_

Round 1

Reviewer 1 Report (Previous Reviewer 3)

The authors addressed my concerns except for one. I still suggest that goal 2 (ii) "to describe changes in these stressors and dietary behaviors over time, from before and during the pandemic" should be perceived changes. A cross-sectional research design cannot describe changes, just perceived ones. Referring to this important point only in the limitation section is not sufficient.

Author Response

The feedback provided by the reviewer helped to significantly improve our manuscript. The reviewer brings up a good point and we have changed the aim to: “(ii) to describe perceived changes in these stressors and dietary behaviors over time, from before and during the pandemic.” We have also updated other sections of the text and the figure accordingly.

Reviewer 2 Report (Previous Reviewer 2)

Dear authors,

I congratulate you on your effort. In my opinion, the material is fine.

Author Response

We appreciate the reviewer taking the time to review our manuscript and for the positive feedback.

This manuscript is a resubmission of an earlier submission. The following is a list of the peer review reports and author responses from that submission.

Round 1

Reviewer 1 Report

Dear authors,

It is an honor for me to review your paper entitled “COVID-19 stressors, psychological distress, and food and beverage consumption among students at a large California state university”.

Your manuscript addresses a useful and necessary topic, at a social, educational, didactic and pedagogical level. Here are some modifications that you must include and without which your paper could not be published.

The keyword COVID-19 stressors must be separated into two different keywords.

The introduction is somewhat superficial. The state of the matter is briefly analyzed but it does not raise the real need for the study nor does it provide in-depth information on the constructs analyzed in the paper (stressors, psychological distress, food and beverage consumption).

Regarding the method, the research design used is not reflected. Is a descriptive study? Exploratory?

The sample does not reflect the sampling error or the inclusion/exclusion criteria of the participants. The ethnic origin of the participants is analyzed. Is culture a variable that is analyzed and on which the analyzed constructs depend? If so, the entire manuscript has to be reworked. Otherwise, there is no coherence with the introduction or with the entire paper.

As for the instruments, the psychometric properties are not specified, nor are their validity and reliability. This has to be reflected.

The ethical considerations and the ethical code followed in the research are not specified.

The limitations must go after the conclusions and include future lines of action.

Reviewer 2 Report

Dear Authors,

Congratulations to the authors for their interesting article! 

The article has considerable potential to contribute to the interrelationships between the stress factors generated by the Covid 19 pandemic and the eating behavior of students. The work addresses a novel theme because the study focuses mainly on the effect of the Covid pandemic on student behavior. Of note is the use of a valuable research instrument, a cross-sectional online survey that was administered to students at California State University.

The authors' analysis shows that the importance of distinguishing between health status and eating habits can be inferred if there are differences by looking at how stressors make their mark. The authors' research is complex and provides a series of recommendations that can be proposed to both educational mentors in universities and medical staff.

The bibliographic references are corresponding to the topic approached in the paper. The others used both references from recent literature and the results of studies relevant to the phenomenon studied. Is the article adequately referenced.

The empirical section is really accurate and very beneficial to all in the academic communities and society.

To help authors improve the level of their article, the following recommendations can be defined:

• The Introduction section should be divided into a section dedicated to the analysis of the current research framework in the field of responsible consumption. It should contain more controversial starting points (opinions, trends) in the field.

• The Conclusions, limitations and future research section should be included and discussed in relation to the results obtained by the authors of the paper. In this part, the authors should write to other researchers how the research is with the competition. What are the similarities and differences.

• Add some social implications to the work.

Reviewer 3 Report

The study examines the associations between COVID-19 stressors, psychological distress, and eating habits among college students in one large state university in the US.

It is based on a cross-sectional research design and many tailored-made questions.

 The combination of measures of psychological distress and dietary behavior is nice and refreshing. The literature review is extensive and updated. However, I have several concerns.

1. The Study aims to examine changes in stressors, distress, and eating habits, while the research design does not allow it. Asking people retrospectively about the changes they underwent is suspected toward students' implicit theories regarding the inherent consistency or change of their attributes, behaviors, and state (Ross, 1989, cited by thousands of papers). Thus, the authors examined students' perceived change rather than actual changes, which should be emphasized throughout the paper. The discussion should expand on this potential bias and accompany their discussion with good literature, including references that found differences between longitudinal studies examining change versus cross-sectional ones that asked about changes.

2. The Study is based on one State university, and it is unclear how it represents other States universities. Authors should relate to this aspect.

3. Authors ignored the disproportion of women in their sample and further indicated in the limitation section that their sample is modestly similar to those of the larger Cal State LA student body" which is not the case for females. This statement should be corrected, and the authors should discuss how this oversampling of women may bias the results.

4. Methods: It is unclear who distributed the link to the students and how their university e-mail confidentiality was kept. 

5. Methods: Authors reported that individuals who did not answer all of the key questions in the survey were excluded. "What were the key questions, and what did the authors mean by "did not respond"? This seems a very harsh method to deal with missing values. What was the rationale for removing a person if they did not respond to one or two questions on a large scale or in one aspect of the study? The authors lost hundred of subjects this way. Also, how did the few hundred that were excluded due to missing values differ in the background or other variables from those who completed the survey?

6. Methods: Authors collapsed categories of many variables (e.g., the worry or financial strain questions) without any explanation for the reason to do so. 
